# Sustainable Insect Pest Management Options for Rice Production in Sub-Saharan Africa

**DOI:** 10.3390/insects16111175

**Published:** 2025-11-18

**Authors:** Esther Pegalepo, Roland Bocco, Geoffrey Onaga, Francis Nwilene, Manuele Tamò, Abou Togola, Sanjay Kumar Katiyar

**Affiliations:** 1M’bé Research Station, Africa Rice Center (AfricaRice), Bouake 01 BP 2551, Côte d’Ivoire; g.onaga@cgiar.org (G.O.); s.katiyar@cgiar.org (S.K.K.); 2Texas A&M AgriLife Research and Extension Center, 1509 Aggie Dr., Beaumont, TX 77713, USA; 3AfricaRice Nigeria Country Office, c/o IITA, Ibadan PMB 5320, Oyo State, Nigeria; f.nwilene@cgiar.org; 4IITA-Benin, Tri Postal, Cotonou 08 BP 0932, Benin; m.tamo@cgiar.org; 5International Maize and Wheat Improvement Center (CIMMYT), ICRAF Campus, United Nations Avenue, Gigiri, Nairobi P.O. Box 1041-00621, Kenya; a.togola@cgiar.org

**Keywords:** sustainable pest management, Sub-Saharan Africa, rice production, integrated pest management (IPM), biopesticides, agricultural sustainability

## Abstract

Major insect pests such as stem borers, leafhoppers, and rice bugs are increasingly threatening rice production in Sub-Saharan Africa (SSA). Sustainable pest management is vital to protect yields and promote agricultural sustainability. Integrated Pest Management (IPM) offers a balanced approach, combining multiple pest management strategies to reduce reliance on chemical inputs and safeguard ecosystems. Key strategies include weed and alternative host plant control, which limits pest habitats, and the use of resistant or tolerant rice varieties that naturally deter insect damage. Biopesticides—derived from natural organisms—are emerging as eco-friendly alternatives to synthetic chemicals. While chemical control remains common, its overuse can harm beneficial insects and the environment. Biological control, using predators and parasitoids, supports long-term pest suppression. A comparison between cultural, chemical, and biological methods demonstrates the value of context-specific solutions tailored to local farming systems. Advances in biotechnology, such as genetic resistance and molecular diagnostics, offer promising tools for future pest management. However, challenges persist, including limited farmer awareness, infrastructure gaps, and climate variability. Future directions should focus on farmer education, policy support, and adaptive research to strengthen sustainable pest management in SSA and ensure resilient rice production systems.

## 1. Introduction

Rice is a crucial staple crop in Sub-Saharan Africa (SSA), playing an important role in food security and economic stability [1]. Insect pests are among the major challenges in rice production in SSA, causing significant yield losses and economic damage [2]. Common pests include the African rice gall midge (*Orseolia oryzivora*, AfRGM), rice stem borers, the grain stink bug (*Aspavia armigera*), and various species of planthoppers and leafhoppers [3,4,5]. These pests can reduce yields by up to 50%, severely impacting food security and farmer incomes [6]. Several factors, such as climatic conditions, cropping practices, and the availability of natural enemies, influence the presence and impact of insect pests [6]. In many regions, the absence of effective pest management tactics and lack of awareness of Integrated Pest Management (IPM) principles and practices exacerbate the problem, leading to recurrent pest outbreaks [6,7]. Additionally, the limited access to pest control resources and extension services among smallholder farmers hinders the adoption of IPM tactics. Addressing these challenges ensures that natural enemies and varietal resistance can be successfully used together during a cropping cycle to bring down pest damage to the crop.

IPM begins with accurate pest identification to learn about its biology and then build a successful management program to control it. This knowledge is essential for developing targeted, effective control strategies. Consistent monitoring of pest populations (e.g., *Spodoptera frugiperda*, *Nymphula depunctallis*, *Trichispa* spp., etc.) and their behaviors is vital for early detection of potential outbreaks and timely intervention to prevent large-scale damage. Furnishing farmers with knowledge, training, and skills in IPM techniques is crucial for successful implementation. IPM emphasizes the use of sustainable practices that minimize environmental impact and promote the long-term health of the agricultural system. Such sustainable practices include the combined use of biological, cultural, and mechanical control methods to minimize pest damage while reducing the reliance on synthetic pesticides [8,9]. Biological control involves the use of natural enemies, such as predators and parasitoids, to manage pest populations, while cultural practices involve the purposeful management of the crop environment to keep pest populations below harmful levels. These practices include crop rotation, intercropping, destruction of alternative host plants, and the use of resistant varieties [9,10]. Mechanical control methods, such as handpicking and the use of traps or barriers, are also employed [11]. However, the use of synthetic pesticides remains common due to their immediate effectiveness [12,13]. Their overuse and misuse have serious consequences, including the development of pesticide-resistant pest populations, environmental contamination, and health risks to farmers and consumers [14,15]. Therefore, promoting IPM practices and reducing the dependence on synthetic pesticides is crucial for sustainable pest management in rice production [16,17].

The purpose of this review is to provide a thorough review of the major insect pests affecting rice production in SSA and examine the various pest management strategies employed, with a particular focus on varietal resistance, cultural practices, biological control, IPM tactics and their effectiveness in reducing pest-related losses. Additionally, the review discusses the implications of chemical pesticide use and underscores the need to implement sustainable and environmentally friendly pest control methods. By addressing these key areas, the review seeks to support the development of effective and sustainable pest management practices, inform future policy formulation, promote the adoption of effective IPM practices, and ultimately contribute to enhance rice productivity and food security in SSA.

## 2. Insect Pests in SSA Rice Production

### 2.1. Common Insect Pests of Rice in SSA

Rice production in SSA is threatened by a wide range of insect pests, which can cause massive yield losses (10–50%), depending on the pest species, crop stage, and management practices [18,19]. Among the most damaging is the African rice gall midge (*Orseolia oryzivora* Harris & Gagné) which induces gall formation on rice plants, stunting growth and reducing yields [20]. Other pests include stem borers, such as the African white borer (*Maliarpha separatella* Ragonot) and the pink borer (*Sesamia calamistis* Hampson), which bore into rice stems, triggering symptoms such as deadhearts and whiteheads, both of which lead to considerable yield reductions [21,22]. Another important pest is *Aspavia armigera* Fabricius, also known as the shield bug (grain stink bug), which affects rice grain quality. Infestations at the milk stage can cause up to 70% loss in grain weight and a significant drop in both paddy and head rice recovery [23]. Sap-sucking insects like the brown planthopper and leafhoppers, such as the green leafhopper (*Nilaparvata lugens* Stal), not only cause direct damage by sucking sap from rice plants but also act as vectors for viruses, such as rice yellow mottle virus and rice tungro virus [22,24].

Defoliators, such as armyworms and cutworms, reduce photosynthetic capacity and overall plant vigor by feeding on foliage [22,25]. The warm and humid climate of many SSA regions favors the rapid multiplication of these pests, often leading to widespread infestations [26,27]. Considering the complex interplay between environmental factors and pest dynamics in the region, effective management of these pests is crucial for ensuring sustainable rice production and food security in SSA. Table 1 shows the most important insect pests that impact rice production in SSA.

### 2.2. Pests Impact on Rice Yield and Quality

Insect pests pose a significant threat to rice production across SSA, with economic losses that can be both extensive and devastating. While average yield reductions range from 10 to 15%, some regions experience losses of up to 90% in severe infestations [6,46,47]. These reductions in yield translate to significant financial losses for farmers who rely on rice not only as a staple crop but also as a source of income [48,49].

Beyond yield losses, insect damage also deteriorates grain quality, decreasing market value and consumer acceptance [50,51]. According to the same authors, pests such as grain stink bugs, stem borers, and planthoppers cause discoloration and broken kernels and reduce nutritional value, all of which lower the commercial appeal of harvested rice.

Several studies across SSA demonstrate the widespread effects of pest infestations and highlight the urgent need for effective management strategies. For instance:

Nigeria and Ghana: The African rice gall midge (*O. oryzivora*) was reported to cause yield losses of up to 60%, leading to severe farm income reductions and increased pest control expenses [52]. Similar infestations were reported in Ghana, underscoring the critical need for improved pest management [53].

Tanzania and Kenya: Stem borers such as *M. separatella* and *S. calamistis* were responsible for yield losses ranging between 10% and 50%, coupled with increased pesticide and labor costs [54]. Research in Kenya found similar results indicating widespread pest impact and reiterated the importance of IPM in mitigating insect pest losses [55].

Benin: Defoliators, including armyworm [56].

Taken together, these case studies underscore the substantial burden insect pests place on rice production and farmer livelihoods in SSA. Table 2 shows the cumulative losses from pest infestations are estimated at USD$3 billion annually for maize, wheat, and rice [57].

Several pest species induce specific damage to rice while it grows. Figure 1 depicts various insects seen in African rice fields and the damage they cause to the fields.

## 3. Pest Management Strategies

### 3.1. Cultural Practices

For centuries, people have used traditional pest management practices to maintain healthy crops and reduce pest populations, often without relying on synthetic chemicals [104]. These methods are sustainable and environmentally friendly, making them valuable components of IPM systems.

#### 3.1.1. Crop Rotation

This method disrupts the life cycles of pests that are specific to certain crops, reducing their populations over time. Evidence of its efficacy was demonstrated in rice, where crop rotation had improved soil health, crop yield, and pest control, using big data [105]. In the context of rice production across Sub-Saharan Africa (SSA), several countries have demonstrated the value of this practice in pest control. In Nigeria, rotating rice with cowpeas significantly reduced the population of rice stem borers [106]. In Tanzania, alternating rice with maize effectively managed rice leafhoppers) [107]. In Uganda, rotating rice with groundnuts helped to control the incidence of rice yellow mottle virus, often spread by insect vectors [107]. In Kenya, rotating rice with soybeans reduced the incidence of rice gall midge [108]. In Ghana, rotating rice with cassava proved successful in managing rice hispa infestations [108]. Beyond pest suppression, crop rotation contributes to soil fertility maintenance and the mitigation of soil-borne diseases [109]. These examples demonstrate how crop rotation is not only a key pest management strategy but also improves the overall agroecosystem resilience.

#### 3.1.2. Intercropping

Intercropping serves as a valuable strategy for pest management by creating a more complex habitat that disrupts pest behavior and reduces their ability to locate their preferred host plants [110]. This practice can also attract beneficial insects, such as predatory insects and parasitoids that prey on pests, thereby providing natural pest control [111]. For instance, intercropping maize with legumes reduces the incidence of stem borers while improving overall crop resilience [112]. Additionally, the diversity of plant species in intercropping systems suppresses weed growth, which in turn reduces habitats that harbor crop pests [113]. In the context of rice, intercropping has been successfully employed as an insect pest management strategy across SSA. In Nigeria, intercropping rice with cowpeas significantly reduced rice stem borer population [114]. According to Himmelstein et al. (2017) [115], the results from a meta-analysis reveal positive impacts of intercropping on crop yield, farmers’ income, and the effect of integrated pest management in Africa. In Uganda, intercropping rice with groundnuts reduced the incidence of rice yellow mottle virus [116]. In Kenya, the practice of intercropping rice with soybeans has reduced the prevalence of rice gall midge [117]. According to Mugisa et al.’s 2000 [118] study on upland rice in Central Uganda, the 4:3 row intercrop ratio is preferred for rice-based intercrops with beans and groundnuts, whereas 3:2 is preferred for rice-maize intercrops. For Daryanto et al. [119], in terms of sustainable agriculture, intercropping can raise soil fertility, provide soil erosion and pest/weed management, and improve soil carbon sequestration without sacrificing land productivity. This understanding is critical given the connections between biotic and abiotic elements in agroecosystems (for example, maximizing sun radiation and periods). Overall, intercropping exemplifies a synergistic approach to sustainable pest management. By naturally lowering pest pressures and encouraging biodiversity, it reduces dependency on synthetic pesticides and enhances agroecosystem resilience. These benefits are especially critical in SSA, where smallholder farmers often have limited access to expensive synthetic inputs.

#### 3.1.3. Manual Removal

Manual pest removal is one of the simplest and most direct methods of pest control. This technique involves physically removing pests from crops by hand or using tools. It is particularly effective for managing large, visible pests such as caterpillars, beetles, and slugs. It is labor-intensive but highly effective in small-scale farming or home gardens [120]. Complementary practices like pruning, weeding, and the destruction of infested plant material further contribute to reducing pest populations and preventing the spread of diseases [121,122]. While not scalable for large commercial farms, manual removal reduces the use of chemical pesticides, which helps to preserve biodiversity, safeguard soil and water quality, and promote healthier ecosystems [123]. Embedding these approaches within broader IPM programs ensures a balance between productivity and sustainability.

#### 3.1.4. Destruction of Alternative Hosts and Volunteer Plants

Managing alternative host plants and weeds (unwanted plants in a specific setting) is a vital cultural strategy in the control of insect pests. Many alternative plant species serve as refuges or breeding grounds for insect pests, maintaining pest populations even in the absence of rice. By reducing or eliminating these alternative host plants, the lifecycle of pests is disrupted, leading to a decrease in their population and reduces the likelihood of damage to rice crops. In addition, by removing the “safe havens” pests rely on outside the main cropping season, farmers can dramatically reducing initial pest pressure. In Africa, notable examples include managing Striga in Kenya, weed suppression in Nigeria, and aquatic weed control in Lake Victoria. Biological strategies in Ethiopia and Ghana further highlight success [124,125]. Table 3 shows a few pests damaging rice production in Africa and their host plants. In Kenya, the International Center on Insect Physiology and Ecology (ICIPE) introduced push-pull technology, which uses a combination of repellent and attractant plants to manage stem borers and striga weeds in rice fields.

#### 3.1.5. Use of Resistant/Tolerant Rice Varieties

The use of plant breeding to develop resistant rice varieties against insect pests is a crucial strategy for sustainable rice production in SSA. By leveraging the genetic diversity found in different rice cultivars, researchers can identify and incorporate resistance genes and quantitative trait loci (QTLs) that confer protection against major insect pests. For instance, the gene *Bph3* provides resistance to the brown planthopper (*N. lugens*) by triggering the production of toxic secondary metabolites that deter feeding [151]. Likewise, *Bph14* confers resistance by reinforcing cell walls and activating defense-related genes [152]. The *Pi9* gene, mainly used to combat rice blast disease (*Pyricularia oryzae*), indirectly helps in reducing the population of stem borers that thrive in weakened plants [153]. Notable regional applications demonstrate the effectiveness of host plant resistance. For instance, the Makassane variety in Mozambique, which carries multiple resistance genes including *Pi9* and *Bph3*, significantly reduces the incidence of both rice blast and brown planthopper infestations [154,155]. Similarly, in Nigeria, the deployment of the NERICA (New Rice for Africa) varieties, which possess QTLs such as *qBph1* and *qBph2*, has improved resistance to the African rice gall midge (*O. oryzivora*) [156]. Adoption of resistant rice varieties has led to a significant reduction in pest infestations, with a reported 30% decrease in yield losses due to pests [26,27,28,29,30,31,32,33,34,35,36,37,38,39,40,41,42,43,44,45,46,47,48,49,50,51,52,53,54,55,56,57,58,59,60,61,62,63,64,65,66,67,68,69,70,71,72,73,74,75,76,77,78,79,80,81,82,83,84,85,86,87,88,89,90,91,92,93,94,95,96,104,105,106,107,108,109,110,111,112,113,114,115,116,117,118,119,120,121,122,123,124,125,126,127,128,129,130,131,132,133,134,135,136,137,138,139,140,141,142,143,144,145,146,147,148,149,150,151,152,153,154,155,156,157]. In Ghana, the use of WITA 4 variety, harboring the *Bph18* gene, has proven effective against the brown planthopper [158]. In addition, adoption of resistant rice varieties, such as NERICA (New Rice for Africa), successfully controlled the African rice gall midge, resulting in higher productivity and reduced crop losses [159]. These examples highlight the importance of using resistant or tolerant rice varieties in IPM in SSA, where environmental challenges and resource constraints limit chemical use. Moreover, the use of plant resistance aligns with climate-smart and sustainable agriculture goals, preserving biodiversity, safeguarding human health, and supporting long-term productivity. Therefore, continued investment in breeding programs and local seed dissemination is essential to expand access to these resistant varieties and strengthen food systems across the region.

Below in Table 4, are a few genes identified in rice for resistance to insect pests and the exhibited resistance mechanisms.

### 3.2. Chemical Control

The use of synthetic pesticides remains one of the most common and effective methods for managing agricultural pests. These substances are specifically formulated to eliminate or deter insects, weeds, and pathogens, thereby protecting crops and enhancing yields. Studies suggest that without pesticide application, global agricultural output could be reduced by over 50% due to pest and disease damage [208]. Consequently, pesticides are considered critical tools for maintaining food security and stabilizing agricultural economies, particularly in regions vulnerable to high pest pressure [209]. In many rice-growing regions of Africa, farmers rely heavily on chemical insecticides to manage major pests such as stem borers, leaf feeders, and plant hoppers. These chemicals are often applied as foliar sprays (contact or systemic) when pest levels reach visible thresholds, or sometimes in calendar-based schedules in absence of monitoring. In West Africa, carbamates (e.g., carbaryl) and pyrethroids (e.g., lambda-cyhalothrin, Deltamethrin) are common [210,211]. Moreover, in Nigeria, the application of chlorpyrifos significantly reduces the population of rice stem borers, leading to increased yields [21]. In East Africa, synthetic pyrethroids and organophosphates are widely used. For instance, in Tanzania, the use of lambda-cyhalothrin has proven effective in managing rice leafhoppers [212]. In Uganda, farmers have successfully used imidacloprid to control the incidence of rice yellow mottle virus, transmitted by insect vectors [36]. In addition, the use of chemical insecticides to control the African rice gall midge led to a 60% reduction in pest populations [213]. In Kenya, the application of carbofuran reduced the prevalence of rice gall midge, resulting in healthier crops [214]. Across several African countries, fipronil has been deployed to manage rice hispa outbreaks [215]. Overall, chemical control offers rapid and cost-effective protection against pest outbreaks, especially during periods of high infestation. Table 5 summarizes a few synthetic pesticides commonly used in SSA rice farms.

### 3.3. Nature-Based Control Options

Nature-based control options, such as biological control, biopesticides, and semi-ochemicals, are characterized by their low impact on non-target organisms, as well as on human, animal, and environmental health.

Biological control is an environmentally sustainable approach that uses natural predators, parasitoids, and beneficial microorganisms to regulate pest populations [262,263]. This method leverages the natural relationships between organisms to reduce pest numbers without relying on chemical pesticides. In SSA, several successful field-level applications demonstrate effectiveness in managing insect pests in rice production. In Nigeria, the introduction of *Trichogramma* wasps, which parasitize rice stem borer eggs, significantly reduced pest populations [38]. In addition, the use of the predatory beetles in the family Coccinellides has proven effective in managing the rice yellow mottle virus, by reducing populations of the aphids vectors, leading to increased yields [264]. In Tanzania, the use of predatory beetles in the family Coccinellidae has helped control rice leafhoppers [265]. In addition, the use of entomopathogenic fungi to control the rice weevil has shown promising results, with a 40% pest mortality [5,266]. In Tanzania, the adoption of IPM strategies, including the use of neem-based biopesticides, has reduced the reliance on synthetic pesticides and improved the sustainability of rice farming [267]. Additionally, in Zambia, the implementation of farmer field schools has empowered local farmers with knowledge and skills to manage pests using environmentally friendly methods. In Uganda and Tanzania, the introduction of pheromone traps has been effective in managing rice stem borer populations, reducing the need for synthetic pesticides and increasing rice [21]. In Uganda, the application of beneficial fungi such as *Beauveria bassiana* has proven successful in managing rice hispa infestations [268]. In West Africa, the release of parasitoid wasps like *Anagrus* spp. helped to limit the prevalence of rice gall midge [269]. In Benin and Côte d’Ivoire, the fungus *Neozygites tanajoa* has effectively managed the cassava green mite, demonstrating its potential for broader pest management applications [270]. In Kenya, the introduction of parasitoid wasps to control the rice yellow stem borer resulted in a 50% reduction in pest populations and a corresponding increase in rice yields [38,271].

By reducing pest populations through natural regulation, these agents help maintain biodiversity, improve crop resilience, and minimize harm to non-target organisms. Continued investment in research, farmer training, and regional adaptation of biocontrol technologies is essential for scaling up their adoption.

Table 6 provides examples of biopesticides, natural predators, parasitoids, and beneficial microorganisms used to control insect pests.

### 3.4. Integrated Farming System

Integrated Farming System (IFS) is an ecologically sustainable approach that integrates crops, livestock, and other agricultural components to create a balanced, self-reliant production system [399]. Its goal is to enhance productivity and profitability while minimizing environmental impact and maximizing the efficient use of available resources [399].

A well-known example is the Integrated Rice-Fish (IRF) System, which has been practiced in Southeast Asia for over 2000 years. This method combines rice cultivation with managed fish farming, creating mutual benefits that improve soil health and reduce pest pressure [400]. This model has been implemented in different agroecological zones in Sub-Saharan Africa [401,402]. The results of the study conducted on IRF in Nigeria showed that it helped to eradicate weeds and harmful insects. The author concluded that Fish in the rice field serves as a biological control to reduce insects and some rice diseases [403].

In Kenya, the incidence of rice stem-borers was significantly reduced by 40% in the IRF system compared to rice monoculture because insect pests, including stem-borers, were eaten by fish [404].

In Liberia, *Tilapia nilotica* (Linnaeus, 1758) was used to significantly control mosquito larvae in the rice field [405], and in Guinea, the IRF system helped to reduce crop pests like caterpillars, rats, and agouti in the field [403,406,407]. It was concluded that this model can use fish as a safe and cheaper alternative to chemical pesticides.

A similar method is rice–duck systems, which utilize ducks to manage weeds and insect pests, effectively boosting rice yields [408]. Additionally, Chinese rice–frog systems, in which frogs serve as biological pest control agents, have helped reduce reliance on chemical fertilizers and pesticides, offering an environmentally sound alternative to conventional farming [409].

By promoting nutrient recycling, natural pest control, and diversified income sources, IFS fosters long-term resilience in agricultural landscapes.

### 3.5. Summary of the Strengths and Limitations of Cultural, Biological, and Chemical Control

Insect pest management in rice production across SSA involves cultural, biological, and chemical control methods, each with distinct advantages and limitations.

Cultural control aims to disrupt pest life cycles and reduce pest populations by altering the environment [410,411]. This method is environmentally sustainable: they reduce reliance on synthetic inputs, supporting soil health and biodiversity. Secondly, it’s cost-effective over time and requires minimal external input once practices are well established. However, the method also presents some limitations, including delayed impact: it often takes multiple seasons before benefits are fully realized and may require significant changes in farming practices [412,413]. Behavioral and systemic barriers involve the requirement for education, training, and long-term adoption by farming communities, and Site-specific effectiveness may be a factor as it may not control pests uniformly across different agroecological zones and may require significant changes in farming practices and longer timeframes to see results.

Biological control methods involve the use of natural enemies, such as predators, parasitoids, and pathogens, to manage pest populations. This method is target-specific: natural enemies focus on specific pests, minimizing harm to beneficial organisms. Once established, populations of natural enemies can regulate pests over time. Moreover, it’s environmentally friendly, posing minimal risk to ecosystems and human health. However, the method also presents some limitations, including climate sensitivity: efficacy can be influenced by temperature, humidity, and habitat conditions [414,415]. It also has scientific complexity as it requires research, monitoring, and technical expertise for successful implementation, and a slower response as this method may not provide immediate pest suppression during outbreaks.

Chemical control methods utilize synthetic insecticides that are fast acting in suppressing pest populations. Advantages include being Fast and effective, as it delivers immediate pest knockdown, especially during high-pressure outbreaks, and widely available as products are accessible and easy to apply with existing tools and predictable results. Farmers often see measurable short-term yield gains. However, the methods present limitations, including resistance development: overuse can lead to resistant pest populations, reducing their long-term efficacy; additionally, it may cause environmental contamination, with risks including water, soil, and air pollution, and non-target effects as it can harm beneficial insects, wildlife, and human health if misused [209].

The optimal pest management strategy depends on site-specific conditions, the pest species involved, farmer resources, and long-term sustainability goals. Integrating these methods through IPM can maximize their strengths and minimize their weaknesses, leading to more effective and sustainable pest management [416,417]. Table 7 highlights the pest management methods. 

### 3.6. Limitations of Conventional and Chemical Methods in Rice Insect Pest Management

Traditional rice insect control methods such as crop rotation, manual removal, and field sanitation often show limited effectiveness when applied in isolation [27]. Crop rotation may fail where alternative hosts of key pests persist nearby or where small landholdings constrain rotation options [418]. Manual removal and handpicking are labor-intensive and impractical for large fields or high pest infestations [419], while cultural practices like flooding or synchronized planting are influenced by water availability, labor, and farmer coordination [420]. Their success also depends on the pest’s biology, environmental conditions, and farmers’ awareness and timing of application. Similarly, reliance on chemical control presents several disadvantages: repeated use can lead to pest resistance, resurgence, and elimination of natural enemies, disrupting ecological balance. the widespread use of chemical pesticides in rice production across SSA results in a range of negative outcomes for human health, animal well-being, and environmental sustainability [421]. Inappropriate handling and overuse of pesticides—often due to limited training and lack of protective equipment—have exposed farmers to acute health issues such as headaches and respiratory disorders [422]. Yarpuz-Bozdogan 2018 [423] revealed that in Cameroon, Ethiopia, Ghana, and Nigeria, almost 33%, 14%, 28%, and 38% of people used masks during pesticide application, with the combined prevalence of glove use being 8%, 12%, 31%, and 51% in those same countries. Furthermore, about 23% of empty pesticide containers were reused in Benin, 26% in Egypt, 31% in Ethiopia, 21% in Ghana, 30% in Nigeria, and 9% in Tanzania. Beyond human exposure, pesticide runoff into nearby water bodies has been shown to harm aquatic ecosystems and reduce biodiversity, affecting fish and other non-target organisms that are vital to rural livelihoods [424]. Additionally, pesticide residues in rice pose food safety risks. Studies have detected organochlorine pesticide residues in locally processed rice in Nigeria, including compounds like endrin and aldrin, which, although within maximum residue limits, still raise concerns about chronic exposure [424]. These residues can bioaccumulate in animals and humans, potentially leading to long-term health effects such as endocrine disruption and cancer [425]. The lack of regulatory enforcement and continued use of unauthorized or highly hazardous pesticides further exacerbate these risks [426]. An integrated pest management (IPM) approach combining cultural, biological, and chemical tools is therefore essential for long-term effectiveness.

## 4. Innovative Approaches and Technologies

### 4.1. Advances in Biotechnology

Biotechnological innovations have revolutionized pest management in rice through tools such as genetic engineering, CRISPR genome editing, and RNA interference (RNAi). These technologies offer precise and durable solutions for managing insect pests, reducing reliance on synthetic pesticides. For instance, the incorporation of the *Bt* gene from *Bacillus thuringiensis* into rice has conferred resistance to stem borers [153]. CRISPR technology has enhanced pest resistance by allowing precise editing of rice genomes to knock out susceptibility genes or introduce resistance traits [187]. Additionally, RNA interference (RNAi) has been employed to silence essential genes in pests, reducing their ability to damage rice plants [427].

### 4.2. Precision Agriculture

Precision agriculture leverages advanced technologies such as sensors, drones, and data analytics to monitor and manage pest populations effectively. Drones with imaging technology are being deployed for rice production in Ghana, where farmer cooperatives are using drones for crop monitoring and input management [427]. Sensors placed in rice fields can detect early signs of pest infestations by monitoring environmental conditions and plant health [428,429]. Drones equipped with high-resolution cameras and multispectral sensors provide real-time aerial imagery, enabling farmers to identify pest hotspots and assess crop health [430]. Data analytics tools process this information to generate actionable insights, allowing for targeted interventions and optimized pesticide use [431]. McCarthy et al., 2023 and Raheem et al., 2021 [432,433] mentioned the use of drones by some African farmers.

### 4.3. Ecological Engineering

Ecological engineering involves habitat manipulation and landscape management to enhance natural pest control. By creating habitats that support natural enemies of pests, such as predators and parasitoids, farmers can reduce pest populations naturally [434]. Key practices include the use of cover crops, maintaining hedgerows, and creating buffer zones that provide refuges and alternative food sources for beneficial insects [435]. Additionally, landscape management practices, such as crop rotation and intercropping, disrupt pest life cycles and reduce their impact on rice crops [111,436]. In Mali, agroecological practices like crop rotation and intercropping with legumes have enhanced pest management by improving soil health and biodiversity, which naturally controls pest populations [437,438]. These case studies highlight the diverse and innovative approaches being used across SSA to tackle pest challenges in rice production. They align seamlessly with the principles of IPM, promoting biodiversity, reducing chemical inputs, and strengthening the resilience of agroecosystems.

## 5. Case Studies and Lessons Learned

Success stories from various countries in SSA offer valuable lessons and best practices for sustainable pest management. Throughout this work, we have highlighted case studies demonstrating the effective use of diverse methods and IPM strategies. Noteworthy examples include:

The push-pull technology was developed by ICIPE in Kenya, which combines trap plants to attract *Spodoptera frugiperda* and repellent intercrops to deter it. This technology is cited as an ideal approach for mixed farming systems, effectively reducing *S. frugiperda* larval density and damage to cereals crops [439].

An important yet previously unmentioned case is the use of farmer field schools in Burkina Faso, Mali, Zambia, Malawi by FAO, and IFAD to empower local farmers with the knowledge and skills needed to manage pests using environmentally friendly methods. This case highlights a critical lesson: community involvement and capacity building are fundamental to the success and sustainability of IPM. Additionally, strong collaboration between research institutions and farming communities, as seen in Nigeria, significantly enhances the adoption and effectiveness of pest management strategies [264]. These case studies emphasize the importance of a holistic and inclusive approach to pest management in SSA. Scaling these lessons across the region has the potential to strengthen low-input, resilient farming systems that uphold both food security and environmental integrity.

## 6. Challenges and Future Directions

### 6.1. Barriers to Adopting Sustainable Practices

Economically, many smallholder farmers lack the financial resources to invest in sustainable technologies and practices, such as biopesticides or pheromone traps [10]. Socially, there is often a lack of awareness and education about sustainable practices, leading to a preference for conventional chemical pesticides [440,441]. Additionally, cultural beliefs and traditional farming practices hinder the acceptance of new methods [33]. On the technical front, several obstacles remain. The availability of sustainable pest control inputs is often restricted, and distribution networks are either underdeveloped or entirely lacking in remote areas. Compounding these issues is the variability of pest pressures and environmental conditions across regions, making it difficult to design and implement one-size-fits-all strategies.

### 6.2. Research Gaps and Future Research Needs

Addressing the challenges of sustainable pest management in SSA requires integrating ecological, socio-economic, and health aspects using frameworks like System Thinking and Digital Twins for dynamic, data-driven pest management. A critical knowledge gap exists due to poor understanding of the factors that trigger pest outbreaks and limited tools for accurate predictions. One critical research gap lies in developing locally adapted pest control solutions that reflect the diverse ecological zones and socio-economic realities across SSA [10]. Tailored approaches are essential to ensure effectiveness and farmer acceptance. Research is also needed to improve the efficacy, affordability, and scalability of biopesticides and other eco-friendly products [440]. Many sustainable products remain out of reach for smallholder farmers due to cost or limited distribution. In Mali, a participatory framework helped farmers prioritize climate-smart IPM innovations across four rice ecologies, enhancing local adaptation and institutional support [442]. AfricaRice’s Participatory Learning and Action Research (PLAR) approach in Nigeria and Côte d’Ivoire empowered farmers to co-develop integrated rice management strategies, improving pest control and productivity in inland valleys [443]. Studies in Uganda showed that training participants effectively transferred IPM knowledge to peers, amplifying adoption and community ownership [444]. There is a need for joint participatory methods in developing and adopting IPM technologies because they foster farmer empowerment, community ownership, greater social acceptance and knowledge-intensive farming systems.

### 6.3. Recommendations and Policy Support Mechanisms

To accelerate the adoption of sustainable pest management practices in SSA, national governments should develop and implement enabling policies and targeted supportive mechanisms. Key recommendations include the following:

Providing financial incentives, such as subsidies for sustainable products, including biopesticides, pheromone traps, and resistant seed varieties, as well as access to low-interest loans. Such measures can help alleviate the economic burden on smallholder farmers [10].

Investment in training and extension services, which is critical to raise awareness and build the capacity of farmers to adopt and implement sustainable techniques effectively [440,445].

Support for research and development. This should be prioritized to produce context-specific, locally adapted pest control solutions and improve the accessibility of sustainable products [33,446].

Regional and international collaboration with research institutions, development agencies, and NGOs. This can facilitate knowledge exchange, innovation, and funding mobilization.

Regulatory frameworks must be strengthened to restrict the use of highly hazardous pesticides and promote safer, more sustainable alternatives.

Policy plays a pivotal role in shaping the trajectory of agricultural innovation. In SSA, where pest pressure is high and farm-level resources are often limited, these recommendations point to a roadmap for transformative change. By lowering financial hurdles and increasing farmer access to information and technologies, policymakers can empower rural communities to transition from chemical-heavy approaches to resilient, low-input farming systems. Moreover, regulation isn’t just about restriction; it’s also about direction. Clear, science-based policies can create an enabling environment where safer products reach the market, research aligns with farmer needs, and innovation is rewarded. Therefore, strengthening the link between public policy, scientific research, and grassroots practice is essential for building an agroecologically sound and food-secure future.

## 7. Conclusions

This review has highlighted various successful pest management practices across different regions in SSA, emphasizing the importance of IPM strategies. Studies from countries such as Kenya, Nigeria, Tanzania, Ghana, Mali, Benin, Uganda, Liberia, Guinea Conakry, Madagascar, and Zambia have demonstrated the effectiveness of diverse approaches. These practices have led to increased rice yields, reduced reliance on synthetic chemicals, and improved sustainability. However, the adoption of these practices faces economic, social, and technical challenges, ranging from limited financial resources and inadequate infrastructure to insufficient farmer awareness and education. Nevertheless, the collective evidence suggests that overcoming these barriers can unlock significant benefits in rice productivity and food security. The success stories highlighted offer a practical framework for designing and scaling effective, locally adapted pest management strategies. Integrating multiple approaches, such as combining biological controls with cultural practices, can improve system resilience and promote long-term sustainability. Looking ahead, it is critical for stakeholders to support the development and dissemination of region-specific pest control solutions. Governments should implement enabling policies that offer financial incentives, access to training, and investment in agricultural extension services. Equally important is the role of collaboration between researchers, international organizations, and local farming communities to foster innovation and knowledge exchange. Finally, strengthening regulatory frameworks to promote sustainable practices and restrict hazardous chemical use is crucial. By adopting a holistic and inclusive approach, SSA can build resilient rice production systems that support both environmental health and rural livelihoods for generations to come.

## Figures and Tables

**Figure 1 insects-16-01175-f001:**
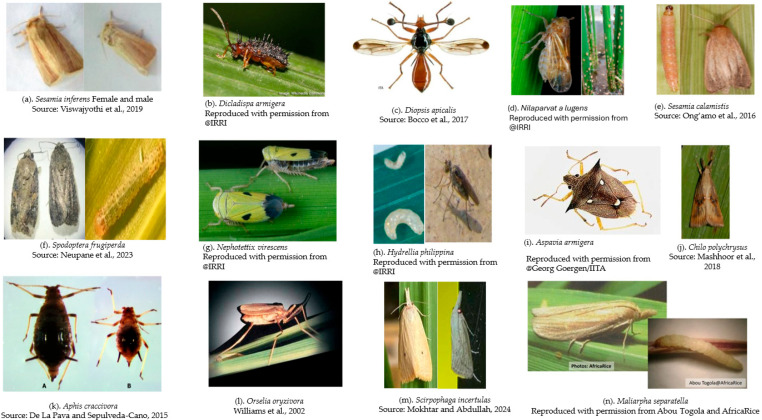
Insect pests of rice plants in SSA [70,97,98,99,100,101,102,103].

**Table 1 insects-16-01175-t001:** List of a few insect pests harmful to rice production in SSA.

Scientific Name	Common Names	Order: Family	Distribution	References
*A. armigera*	Shield Bug or grain stink bug	Hemiptera: Pentatomidae	Across SSA	[23,28,29]
*Chilo zacconius* Bleszynski	Striped stem borer	Lepidoptera: Crambidae	Benin, Cameroon, Côte d’Ivoire, Mali, Niger, Nigeria, Senegal, Sierra Leone	[30,31,32]
*Diopsis longicornis* Macquart, *Diopsis apicalis* Dalman, *Diopsis collaris* Westwood	Stalk-eyed flies	Diptera: Diopsidae	Benin, BurkinaFaso, Cameroon, Chad, Côte d’Ivoire, Gambia, Ghana, Guinea, Guinea-Bissau, Liberia, Mali, Mauritania, Niger, Nigeria, Senegal, Sierra Leone, Togo	[32,33]
*Eldana saccharina*	African sugarcane stalk borer	Pyralidae	Across SSA	[34,35]
*M. separatella* Ragonot	African white borer	Lepidoptera: Pyralidae	Côte d’Ivoire, Mali, Nigeria	[22,32]
*O. oryzivora*	African rice gall midge	Cecidomyiidae	Across SSA	[20,26,31]
*Rhopalosiphum rufiabdominalis*	African rice root aphid	Aphididae	Across SSA	[3,36]
*Scirpophaga melanoclista* Meyrick	Yellow stem borer	Lepidoptera: Crambidae	Cameroon, Côte d’Ivoire, Mali, Nigeria, Senegal	[32,37,38,39]
*S. subumbrosa* Meyrick	Yellow stem borer	Lepidoptera: Crambidae	Ghana, Mali	[32]
*Sesamia calamistis* Hampson	Pink stalk borer	Lepidoptera: Noctuidae	Cameroon, The Gambia, Ghana, Côte d’Ivoire, Niger, Nigeria	[22,32]
*Sesamia nonagriodes botanephaga* Tams & Bowden	Pink stalk borer	Lepidoptera: Noctuidae	Ghana, Côte d’Ivoire, Nigeria	[40,41]
*Sesamia n. penniseti* Tams and Bowden	Pink stalk borer	Lepidoptera: Noctuidae	Ghana, Côte d’Ivoire, Nigeria	[32,42]
*Sesamia poephaga* Tams and Bowden	Pink stalk borer	Lepidoptera: Noctuidae	Nigeria	[32]
*Spodoptera frugiperda*	Fall armyworm	Noctuidae	Across SSA	[43,44,45]

**Table 2 insects-16-01175-t002:** Summary of insect pests and transmitted diseases affecting rice production in SSA and their associated losses.

Target Insects	Damage	Affected Stages	Yield Losses	References
Aphids *Aphis craccivora* *A. gossypii* *Myzus persicae*	Sucking sap, causing yellowing and stunted growth		From 20% to 80%	[58,59]
Shield Bug or grain stink bug *A. armigera*	Sucking sap from grains, causing grain weight loss and quality degradation	Causes the most significant damage to rice during the milk stage, where it sucks sap from developing grains, leading to substantial yield and quality losses	Up to 70%	[60,61]
Dark-headed stem borer *C. polychrysus*	Boring into stems, causing deadhearts and whiteheads		Up to 30%	[62,63,64]
Striped stem borer *C. zacconius* *C. suppressalis*	Boring into stems, causing deadhearts and whiteheads	Primarily affects rice during the tillering and booting stages.	Up to 30%%	[62,65]
Rice hispa *Dicladispa armigera*	Leaf scraping, causing reduced photosynthesis	Primarily affects rice during the vegetative stage, which includes the seedling and tillering stages.	From 10% to 62%	[66,67,68]
Stalk-eyed flies *D. longicornis* *D. apicalis* *D. collaris*	Feeding on rice plants, causing stunted growth and reduced yield	Primarily affects rice during the tillering and flowering stages.	From 10% to 15%	[21,33,69,70]
African sugarcane stalk borer *E. saccharina*	Boring into stems, causing deadhearts and whiteheads	During the late growth stages, particularly during the grain filling stage.	Up to 50%	[71,72]
Rice whorl maggot *Hydrellia philippina*	Feeding on young leaves, causing stunted growth		From 20% to 30%	[73,74]
African white borer *M. separatella* Ragonot	Boring into stems, causing deadhearts and whiteheads		From 34.3% to 90.9%	[75,76]
Rice armyworm *Mythimna separata*	Defoliation, feeding on leaves and stems		From 3% to 70%	[77,78]
Rice leafhopper *Nephotettix virescens*	Sucking sap, causing yellowing and stunted growth		Up to 60%	[79,80]
Brown planthopper *Nilaparvata lugens*	Hopper burn, wilting, plant death	Primarily affects rice during the tillering to booting stages.	From 25% to 60%	[81,82,83]
African rice gall midge *O. oryzivora*	Gall formation, stunted growth, reduced tillering	Primarily affects rice during the vegetative stage, which includes the seedling and tillering stages.	Up to 100%	[20,84,85]
Pink stalk borer *S. calamistis* *S. nonagriodes botanephaga* *S. n. penniseti* *S. poephaga* *S. inferens*	The larvae tunnel into the stems, causing significant damage such as “dead hearts” (where the central shoot dies), broken stems, and reduced yield.	Early growth stages, including the seedling and early tillering	From 25.7% to 78.9%	[86,87,88]
Yellow stem borer *S. incertulas*	Boring into stems, causing deadhearts and whiteheads	It affects rice from the seedling stage through to maturity.	From 3% to 87%	[87,89,90]
Fall Armyworm *S. frugiperda*	Defoliation, feeding on leaves and stems		From 10% to 73%	[91,92,93]
White Stem Borer *S. innotata*	Boring into stems, causing deadhearts and whiteheads	Primarily affects rice during the tillering and booting stages.	Up to 80%	[94,95,96]

**Table 3 insects-16-01175-t003:** Rice Insect Pests in SSA and their Common Alternate Host Plants.

Insect Names	Weed Hosts (Common and Scientific Names)	References
*C. zacconius* Bleszynski	Broadleaf Signalgrass (*B. platyphylla*), Crabgrass (*Digitaria* spp.)	[126,127]
*D. longicornis* Macquart, *D. apicalis* Dalman, *D. collaris* Westwood	Goosegrass (*E. indica*), Johnsongrass (*S. halepense*)	[128,129]
*E. saccharina*	Barnyardgrass (*E. crus-galli*), Yellow Nutsedge (*Cyperus esculentus*)	[130,131,132]
*M. separatella* Ragonot	Broadleaf Signalgrass (*B. platyphylla*), Crabgrass (*Digitaria* spp.)	[133,134]
*O. oryzivora*	Barnyardgrass (*Echinochloa crus-galli*), Yellow Nutsedge (*Cyperus esculentus*)	[26,135]
*R. rufiabdominalis*	Goosegrass (*Eleusine indica*), Johnsongrass (*Sorghum halepense*)	[131,136]
*S. melanoclista* Meyrick	Goosegrass (*E. indica*), Johnsongrass (*Sorghum halepense*)	[56,137]
*S. subumbrosa* Meyrick	Barnyardgrass (*E. crus-galli*), Yellow Nutsedge (*Cyperus esculentus*)	[138,139]
*S. calamistis* Hampson	Barnyardgrass (*Echinochloa crus-galli*), Johnsongrass (*Sorghum halepense*)	[131,140]
*S. nonagriodes botanephaga* Tams & Bowden	Barnyardgrass (*E. crus-galli*), Yellow Nutsedge (*Cyperus esculentus*)	[82,141]
*S. n. penniseti* Tams and Bowden	Broadleaf Signalgrass (*Brachiaria platyphylla*), Crabgrass (*Digitaria* spp.)	[142,143,144]
*S. poephaga* Tams and Bowden	Goosegrass (*Eleusine indica*), Johnsongrass (*Sorghum halepense*)	[145,146,147]
*S. frugiperda*	Broadleaf Signalgrass (*Brachiaria platyphylla*), Crabgrass (*Digitaria* spp.)	[148,149,150]

**Table 4 insects-16-01175-t004:** List of a few genes/*QTLs* identified in rice to control insects along with resistance mechanism and donors.

Target Insects	Resistance Sources	Genes/QTLs/Resistance Mechanism	References
*A. craccivora*	IR64	*Rag1*	[160]
*A. gossypii*	IR36	Cucurbitacin C	[161]
*C. polychrysus* (Dark-headed stem borer)	IR36	Antibiosis: phenolic compounds affect the growth and development of larvae.	[64]
	IR50	Antixenosis: secondary metabolites like catechetic tannins reduce the plant attractiveness.	[162]
	IR13429-57-1	Antibiosis: flavonoid and other biochemical compounds affect larval development and survival.	[163]
*C. zacconius* *C. suppressalis* (Striped stem borer)	Lac 23	Antibiosis: phenolic compounds affect the growth and development of larvae.	[164]
	IR 2035-120-3	Antixenosis: secondary metabolites like catechetic tannins reduce the plant attractiveness.	[162]
	IR 4625-132-1-2	Antibiosis: flavonoid and other biochemical compounds affect larval development and survival.	[165]
*D. armigera* (Rice hispa)	Naggar Dhan	Antibiosis: phenolic compounds affect the growth and development of larvae.	[166]
	HPR 2617	Antixenosis: secondary metabolites like catechetic tannins reduce the plant attractiveness.	[166]
	Sukara Dhan	Antibiosis: flavonoid and other biochemical compounds affect larval development and survival.	[166,167]
*D. longicornis* *D. apicalis* *D. collaris* (Stalk-eyed flies)	NERICA4	Antibiosis: phenolic compounds affect the growth and development of larvae.	[168]
	NERICA1 and CG14	Antixenosis: secondary metabolites like catechetic tannins reduce the plant attractiveness.	[33,169]
	NERICA16	Antibiosis: flavonoid and other biochemical compounds affect larval development and survival.	[170]
*E. saccharina* (African sugarcane stalk borer)	WAB56-104	Antibiosis: phenolic compounds affect growth and development.	[171]
	CG14	Antixenosis: tannin and other biochemical compounds reduce the plant attractiveness.	[172]
	ITA306	Antibiosis: flavonoid and other biochemical compounds affect larval development and survival.	[172]
*H. philippina* (Rice whorl maggot)	Swarna-Sub 1	Antibiosis: flavonoid and other biochemical compounds affect larval development and survival.	[173,174]
*M. separata* (Rice armyworm)	IR36, IR64, TNAU Rice, and ADT 37	Antibiosis: affects insect development and survival	[175,176,177]
*M. separatella* Ragonot (African white borer)	BG 90-2	Antibiosis: phenolic compounds affect the growth and development of larvae.	[30]
	Basmati 217	Antixenosis: secondary metabolites like catechetic tannins reduce the plant attractiveness.	[178]
	M27615	Antibiosis: flavonoid and other biochemical compounds affect larval development and survival.	[179]
*M. persicae*	IR72	*Mi-1.2*	[180]
*N. virescens* (Rice leafhopper)	PTB33	*Grh2*. Antibiosis: affects the growth and development of nymphs and adults, leading to high mortality.	[181,182]
	IR64	*Grh4*. Antixenosis: reduces the attractiveness of the plant to the pest, leading to fewer eggs laid and lower survival rates.	[181]
	APL 796	Antibiosis: phenolic compounds affect the growth and development of larvae.	[183]
*N. lugens* (Brown planthopper)	Mudgo	*Bph1*	[184]
	ASD7	*Bph2*	[185]
	Rathu Heenati	*Bph3*	[151]
	Babawee	*Bph4*	[186]
	ARC 10550	*Bph5*	[187]
	Swarnalata	*Bph6*	[185]
	Balamawee	*Bph9*	[188]
	IR65482-7-216-1-2	*Bph10*	[186,189]
	B5	*Bph14*	[190,191]
	IR65482-7-216-1-2	*Bph18*	[186]
*O. oryzivora* (African rice gall midge)	TOG7106	Antixenosis: secondary metabolites like catechetic tannins reduce the plant attractiveness.	[192,193]
	TOS14519	Antibiosis: flavonoid and other biochemical compounds affect larval development and survival.	[33,193]
*S. calamistis* *S. nonagriodes botanephaga* *S. n. penniseti* *S poephaga*, *S. inferens* (Pink stalk borer)	WAB56-104	Antibiosis: phenolic compounds affect growth and development.	[194]
	CG14	Antixenosis: tannin and other biochemical compounds reduce the plant attractiveness.	[194]
	ITA306	Antibiosis: flavonoid and other biochemical compounds affect larval development and survival.	[195,196,197,198]
*S. frugiperda* (Fall Armyworm)	Miúdo Branco	Antixenosis: non-preference by insects	[199]
	IR 64	Antibiosis: affects insect development and prolongs life cycle.	[199]
	Bacaba Branco	Antibiosis: affects insect development and reduces nutritional indices.	[199]
*S. incertulas* (Yellow stem borer)	TKM6	Antibiosis: phenolic compounds affect the growth and development of larvae.	[200,201]
	IR36	Antixenosis: secondary metabolites like catechetic tannins reduce the plant attractiveness.	[202,203]
	PTB33	Antibiosis: flavonoid and other biochemical compounds affect larval development and survival.	[204,205]
*S. innotata* (White Stem Borer)	KSK-456	Antibiosis: phenolic compounds affect the growth and development of larvae.	[95]
	PK 9586-8-2	Antixenosis: secondary metabolites like catechetic tannins reduce the plant attractiveness.	[206]
	BRRI Dhan 64	Antibiosis: flavonoid and other biochemical compounds affect larval development and survival.	[207]

**Table 5 insects-16-01175-t005:** List of Synthetic Pesticides used to Control Pests in SSA Rice Farms.

Names	Commercial Name	Active Ingredients	Dose and Application	References
*C. zacconius* Bleszynski (Striped stem borer)	Regent, Karate, Bulldock, Virtako	Fipronil, Lambda-cyhalothrin, Beta-cyfluthrin, Chlorantraniliprole	0.2 L/ha, 0.5 L/ha, 0.3 L/ha, 0.15 L/ha, foliar spray	[21,216,217,218]
*D. longicornis* Macquart *D. apicalis* Dalman *D. collaris* Westwood (Stalk-eyed flies)	Karate, Bulldock, Virtako, Confidor	Lambda-cyhalothrin, Beta-cyfluthrin, Chlorantraniliprole, Imidacloprid	0.5 L/ha, 0.3 L/ha, 0.15 L/ha, 0.25 L/ha, foliar spray	[115,219,220,221,222]
*E. saccharina* (African sugarcane stalk borer)	Bulldock, Karate, Virtako, Confidor	Beta-cyfluthrin, Lambda-cyhalothrin, Chlorantraniliprole, Imidacloprid	0.3 L/ha, 0.5 L/ha, 0.15 L/ha, 0.25 L/ha, foliar spray	[5,216,223,224,225,226,227,228]
*M. separatella* Ragonot (African white borer)	Karate, Bulldock, Virtako, Confidor	Lambda-cyhalothrin, Beta-cyfluthrin, Chlorantraniliprole, Imidacloprid	0.5 L/ha, 0.3 L/ha, 0.15 L/ha, 0.25 L/ha, foliar spray	[115,216,229,230,231,232]
*O. oryzivora* (African rice gall midge)	Karate, Bulldock, Virtako, Confidor	Lambda-cyhalothrin, Beta-cyfluthrin, Chlorantraniliprole, Imidacloprid	0.5 L/ha, 0.3 L/ha, 0.15 L/ha, 0.25 L/ha, foliar spray	[115,216,233,234,235,236]
*R. rufiabdominalis* (African rice root aphid)	Confidor, Karate, Bulldock, Virtako	Imidacloprid, Lambda-cyhalothrin, Beta-cyfluthrin, Chlorantraniliprole	0.25 L/ha, 0.5 L/ha, 0.3 L/ha, 0.15 L/ha, foliar spray	[216,237,238,239,240,241,242,243]
*S. melanoclista* Meyrick (Yellow stem borer)	Fipronil Cartap Hydrochloride	Fipronil Cartap Hydrochloride	0.3 GR at 2.5 g/m^2^ 4% GR at 1.9 g/m^2^	[244,245,246]
*S. subumbrosa* Meyrick (Yellow stem borer)	Chlorpyriphos	Chlorpyriphos 75 WDG	500–533 g/ha	[207]
*S. calamistis.* Hampson *S. nonagriodes botanephaga* Tams & Bowden *S. n. penniseti* Tams and Bowden *S. poephaga* Tams and Bowden (Pink stalk borer)	Karate, Bulldock, Virtako, Confidor	Lambda-cyhalothrin, Beta-cyfluthrin, Chlorantraniliprole, Imidacloprid	0.5 L/ha, 0.3 L/ha, 0.15 L/ha, 0.25 L/ha, foliar spray	[5,216,234,247,248,249,250,251,252]
*S. frugiperda* (Fall armyworm)	Belt, Karate, Bulldock, Virtako	Flubendiamide, Lambda-cyhalothrin, Beta-cyfluthrin, Chlorantraniliprole	0.2 L/ha, 0.5 L/ha, 0.3 L/ha, 0.15 L/ha, foliar spray	[216,253,254,255,256,257,258,259,260,261]

**Table 6 insects-16-01175-t006:** List of selected nature-based control options.

Scientific Name	Common Names	Nature-Based Control Options	References
*C. polychrysus*	Dark-headed stem borer	*Cotesia flavipes*: A parasitoid wasp that targets the larvae. *Trichogramma chilonis*: An egg parasitoid that targets the eggs. *Beauveria bassiana*: An entomopathogenic fungus used for controlling the larvae. *Bacillus thuringiensis* (Bt): A microbial insecticide effective against the larvae. Neem-based products: Neem oil and neem cake have been used. Chlorantraniliprole nano-pesticides: Eco-friendly chitosan-based formulations for effective control.	[162,272,273]
*C. suppressalis*	Striped stem borer	*Trichogramma japonicum*: An egg parasitoid that targets the eggs. *Cotesia flavipes*: A parasitoid wasp that targets the larvae. *Beauveria bassiana*: An entomopathogenic fungus used for controlling the larvae. *Bacillus thuringiensis* (Bt): A microbial insecticide effective against the larvae. Neem-based products: Neem oil and neem cake have been used. Chlorantraniliprole nano-pesticides: Eco-friendly chitosan-based formulations for effective control.	[274,275,276,277,278]
*C. zacconius* Bleszynski	Striped stem borer	*Cotesia flavipes*: A parasitoid wasp that targets the larvae. *Trichogramma chilonis*: An egg parasitoid that targets the eggs *Beauveria bassiana*: An entomopathogenic fungus used for controlling the larvae. *Bacillus thuringiensis* (Bt): A microbial insecticide effective against the larvae. Neem-based products: Neem oil and neem cake have been used.	[38,115,279,280,281]
*D. armigera*	Rice hispa	*Trichogramma zahiri*: An egg parasitoid wasp that targets the eggs. *Neochrysocharis spp*.: An egg and larval parasitoid effective. *Scutibracon hispae*: A larval and pupal parasitoid. Neem-based products: Neem oil and neem cake have been used. Azacel: A commercial biopesticide that has shown high efficacy. Larvocel: Another commercial biopesticide effective in reducing the population of *Dicladispa armigera*.	[282,283,284,285,286,287,288]
*D. longicornis* Macquart *D. apicalis* Dalman *D. collaris* Westwood	Stalk-eyed flies	*Trichogramma chilonis*: An egg parasitoid that targets the eggs. *Cotesia flavipes*: A parasitoid wasp that targets the larvae. Neem-based products: Neem oil and neem cake have been used.	[289,290,291]
*E. saccharina*	African sugarcane stalk borer	*Cotesia flavipes*: A parasitoid wasp that targets the larvae. *Trichogramma chilonis*: An egg parasitoid that targets the eggs. Beauveria bassiana: An entomopathogenic fungus used for controlling the larvae. *Beauveria bassiana*: An entomopathogenic fungus used for controlling the larvae. ASTOUN 50 EC: A biopesticide that has shown antiappetizing and repellent effects. NECO 50 EC: Another biopesticide tested for its effects.	[292,293,294,295,296]
*H. philippina*	Rice whorl maggot	*Trichogramma spp*.: Egg parasitoids that target the eggs. *Cotesia flavipes*: A parasitoid wasp that targets the larvae. *Beauveria bassiana*: An entomopathogenic fungus used for controlling the larvae. Neem-based products: Neem oil and neem cake have been used. Flubendiamide: A biopesticide effective against the larvae. Spinosad: Another biopesticide that has shown efficacy against the larvae.	[90,297,298,299,300,301,302,303,304,305]
*M. separatella* Ragonot	African white borer	*Chelonus maudae*: A parasitoid wasp that targets the larvae. *Rhaconotus carinatus*: Another parasitoid wasp effective against the larvae. *Pristomerus bullis*: An ichneumonid wasp that parasitizes the larvae. *Bacillus thuringiensis* (Bt): A microbial insecticide effective against the larvae. Neem-based products: Neem oil and neem cake have been used.	[21,30,37,117,178,301,306,307,308]
*M. separata*	Rice armyworm	*Cotesia ruficrus*: A parasitoid wasp that targets the larvae. *Trichogramma chilonis*: An egg parasitoid effective. *Trichogramma dendrolimi*: Another egg parasitoid used. Nomuraea rileyi: An entomopathogenic fungus that infects and kills the larvae. *Bacillus thuringiensis* (Bt): A bacterial biopesticide that produces toxins specifically targeting the larvae.	[309,310,311,312,313,314,315,316,317]
*N. virescens*	Rice leafhopper	*Trichogramma japonicum*: An egg parasitoid that targets the eggs. *Anagrus spp*.: Egg parasitoids that are effective against the egg. *Beauveria bassiana*: An entomopathogenic fungus used for controlling the nymphs and adults. Neem-based products: Neem oil and neem cake have been used. *Bacillus thuringiensis* (Bt): A microbial insecticide effective against the nymphs and adults.	[216,318,319,320,321,322,323,324,325]
*N. lugens*	Brown planthopper	*Beauveria bassiana*: An entomopathogenic fungus used for controlling the nymphs and adults. *Lecanicillium attenuatum*: Another entomopathogenic fungus that has shown significant control efficacy. *Trichogramma japonicum*: An egg parasitoid that targets the eggs. Neem-based products: Neem oil and neem cake have been used. *Bacillus thuringiensis* (Bt): A microbial insecticide effective against the nymphs and adults.	[165,186,216,286,326,327,328,329,330,331,332]
*N. depunctalis*	Rice caseworm	Snails: Feed on the eggs. Hydrophilid and Dytiscid water beetles: Feed on the larvae. Spiders, Dragonflies, and Birds: Predate on the adult caseworms. Parasitoids: Such as *Elasmus* spp., *Apanteles* spp., *Bracon* spp., and *Pediobius* spp. Nuclear Polyhedrosis Virus: A potential pathogen for controlling *N. depunctalis*. Plant Extracts: Extracts from *Calotropis procera* and *Zanthoxylum nitidum* have demonstrated significant insecticidal properties against *N. depunctalis* larvae.	[333,334,335,336,337,338,339,340,341]
*O. oryzivora*	African rice gall midge	*Platygaster diplosisae*: A parasitoid wasp that targets the larvae. *Aprostocetus procereae*: Another parasitoid wasp effective against the larvae. *Metarhizium anisopliae*: An entomopathogenic fungus used for controlling the larvae. *Beauveria bassiana*: Another entomopathogenic fungus effective against the larvae. Neem-based products: Neem oil and neem cake have been used. Eucalyptus extracts: Effective in reducing the incidence of galls caused by the larvae.	[52,85,342,343,344,345]
*R. rufiabdominalis*	African rice root aphid	*Stratiolaelaps scimitus* (syn. *Hypoaspis miles*): A soil-dwelling predatory mite that targets the rice root aphid. *Beauveria bassiana*: An entomopathogenic fungus that infects and kills the aphids. *Verticillium lecanii* (now known as *Lecanicillium lecanii*): Another entomopathogenic fungus effective against aphids. *Beauveria bassiana* (e.g., BotaniGard 22 WP, Mycotrol WPO): These biopesticides are applied to control root aphid populations by infecting and killing them.	[242,346,347,348,349,350,351,352]
*S. incertulas*	Yellow stem borer	*Trichogramma japonicum*: An egg parasitoid that targets the eggs. *Cotesia flavipes*: A parasitoid wasp that targets the larvae. *Telenomus rowani*: Another egg parasitoid effective against the eggs. *Bacillus thuringiensis* (Bt): A microbial insecticide effective against the larvae. Neem-based products: Neem oil and neem cake have been used. Lemongrass oil: Effective in reducing the incidence of white ear heads.	[280,286,353,354,355,356,357,358,359,360]
*S. innotata*	White Stem Borer	*Trichogramma japonicum*: An egg parasitoid that targets the eggs. *Cotesia flavipes*: A parasitoid wasp that targets the larvae. *Beauveria bassiana*: An entomopathogenic fungus used for controlling the larvae. *Bacillus thuringiensis* (Bt): A microbial insecticide effective against the larvae. Neem-based products: Neem oil and neem cake have been used. Eucalyptus oil: Effective in reducing the incidence of white ear heads.	[361,362,363,364,365,366]
*S. melanoclista* Meyrick *S. subumbrosa* Meyrick	Yellow stem borer	*Trichogramma japonicum*: An egg parasitoid that targets the eggs. *Cotesia flavipes*: A parasitoid wasp that targets the larvae. *Beauveria bassiana*: An entomopathogenic fungus used for controlling the larvae. *Bacillus thuringiensis* (Bt): A microbial insecticide effective against the larvae. Neem-based products: Neem oil and neem cake have been used. Chlorantraniliprole nano-pesticides: Eco-friendly chitosan-based formulations for effective control.	[37,95,217,355,367]
*S. calamistis* Hampson	Pink stalk borer	*Cotesia sesamiae*: A parasitoid wasp that targets the larvae of *Sesamia calamistis.* *Metarhizium anisopliae*: An entomopathogenic fungus effective against the larvae. *Beauveria bassiana*: Another entomopathogenic fungus used for controlling the larvae. Neem-based products: Neem oil and neem cake have been used.	[269,368,369,370,371,372]
*Sesamia inferens*	Pink stem borer	*Cotesia flavipes*: A parasitoid wasp that targets the larvae. *Tetrastichus howardi*: A parasitoid wasp effective against the larvae. *Trichogramma* spp.: Egg parasitoids that target the eggs. *Bacillus thuringiensis* (Bt): A microbial insecticide effective against the larvae. *Beauveria bassiana*: An entomopathogenic fungus used for controlling the larvae. Neem-based products: Neem oil and neem cake have been used.	[261,373,374,375,376,377]
*S. nonagriodes botanephaga* Tams & Bowden	Pink stalk borer	*Trichogramma* spp.: Egg parasitoids that target the. *Cotesia flavipes*: A parasitoid wasp that targets the larvae. *Bacillus thuringiensis* (Bt): A microbial insecticide effective against the larvae.	[214,370,378,379,380]
*S. n. penniseti* Tams and Bowden	Pink stalk borer	*Cotesia sesamiae*: Effective against the larvae. *Beauveria bassiana*: Used to control the larvae. Neem-based products: Effective in managing.	[381,382,383]
*S. poephaga* Tams and Bowden	Pink stalk borer	*Cotesia sesamiae*: Effective against the larvae. *Cotesia flavipes*: A parasitoid wasp that targets the larvae. *Bacillus thuringiensis* (Bt): A microbial insecticide effective against the larvae.	[384,385,386,387,388]
*S. frugiperda*	Fall Armyworm	*Telenomus remus*: A parasitoid wasp that targets the eggs. *Trichogramma pretiosum*: Another egg effective parasitoid. *Chelonus insularis*: A larval parasitoid that attacks the early stages of the pest. *Steinernema riobrave*: An entomopathogenic nematode that infects and kills the larvae. *Metarhizium anisopliae*: An entomopathogenic fungus effective against various stages of the pest. *Bacillus thuringiensis* var. *kurstaki* (Bt): A bacterial biopesticide that produces toxins specifically targeting the larvae.	[389,390,391,392,393,394,395,396,397,398]

**Table 7 insects-16-01175-t007:** Comparison between cultural, chemical, and biological control methods.

Comparison	Cultural Method	Chemical Method	Biological Method
Common		Aim to control pests and diseases	
		Improve crop yield	
	Manual practices	Use of synthetic chemicals	Use of natural predators/pathogens
	Crop rotation	Pesticides and herbicides	Biopesticides and beneficial insects
Differences	Water management	Quick action	Environmentally friendly
	Labor-intensive	Potential resistance development	Sustainable
	Lower cost	Higher cost	Moderate cost

## Data Availability

No new data were created or analyzed in this study. Data sharing is not applicable to this article.

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
