# Peer review of "Sustainable Insect Pest Management Options for Rice Production in Sub-Saharan Africa"

_insects, 2025, doi:10.3390/insects16111175_

Round 1

Reviewer 1 Report

Comments and Suggestions for Authors

This review summarizes and complies the occurrence and prevention of rice insect pests in sub-Saharan Africa. This summary is of great significance for fully understanding the current situation and guiding future prevention and control on rice insect pests in sub-Saharan Africa. I suggest minor revision for the manuscript. Some revisions are needed before acceptance, which are listed below:

  1. Line 112, The bracket is incomplete.
  2. Line 139, should revise to “Insect pests on rice plant in SSA”
  3. In figure 1, some photos are not clear, this need to be revised. And, the text “male” and “female” should move to the photo.
  4. Line 209, line 265, line 306, the font format is incorrect.
  5. Line 243, “3.2.1. Chemical pesticides used” should be deleted.
  6. Figure 4, the resolution of the image is too low.
  7. In all Tables, the order of insect species should be arranged alphabetically. And, the format of the tables should be standardized.
  8. In section 6 (line 468), more information about good practices and methods in other countries should be added.
  9. The manuscript contains a lot of contents and writing errors must be carefully checked.

Author Response

Dear Reviewer 1, 

Hope you are doing well? 

Many thanks for your comments that help us to improve the manuscript.

Find attached the point-by -point answers to your comments.

Best regards

Esther 

Reviewer 2 Report

Comments and Suggestions for Authors

Review of sustainable pest management rice in Africa

Line 42 – there must be citations for the first sentence about rice as a crop.  If the authors use the term significant, then there needs to be sources to back up and provide data for why it is significant. 

Line 43 – Since this is a review and the authors are trying to establish the importance of insects as pests of rice, there needs to be a bit more explanation about it.  If insects are one of the major challenges, what are some examples of the other challenges?  If insects aren’t the most important challenge, then how do they rank?  Are they the second or third most important challenge? 

Line 48 – The term “prevalence” is confusing.  Do the authors mean how common the entire group of rice insect pests are in the entire area of SSA?  Prevalence is expressed as a percentage or rate out of some larger population or sample.  It could also have a time or seasonal factor.  High or low prevalence equates to widespread or rare, but that’s difficult to measure over such a vast area.  Do the authors mean localized prevalence?  In any case, insect species diversity – the number of different species in any given area and a species’ population abundance in any given area are influenced by an extraordinary number of factors beyond climate and cropping practices.  Things like commerce and artificial movement of species into an area, the floristic composition of an area, etc.  For any given farm, the types of insect pests and their abundances are influenced by surrounding farms, surrounding vegetation, weather and microclimates, and pest management practices in other managed areas.  The one thing that likely has little effect on the types of insect species over a wide area is availability of natural enemies.  Availability of natural enemies may not even have any impact on prevalence (the amount) of any given insect pest over a localized farmscape, let alone all of SSA, because availability doesn’t automatically translate to reductions in pest prevalence – in other words it is not a given. 

Line 49 – Are the authors referring to insect pests challenging rice production as the problem being exacerbated by a lack of effective pest management strategies, etc.?  The lack of awareness (by growers, I assume) of IPM principles and practices do not lead to recurrent pest outbreaks.  There needs to be a separation of pest management strategies and IPM.  In fact, I would use the term pest management tactics.  And how you group and use pest management tactics is the strategy.  IPM is not a pest management tactic.  IPM is an organizing principle – a strategy - for pest management tactics.  If there are pest management tactics that are missing or can’t be implemented, then the strategy is affected, and overall efficacy is compromised.  Having a lack of awareness of IPM is not that bad of an issue.  Growers will use whatever pest management tactics they have access to and can afford to try and manage insect pests.  The main question to ask is – are growers being thoughtful about using insecticides and natural enemies at the same time?  Because that is the crux of integrated pest management.  Those are the two tactics that need careful integration.  They are the only factors that need careful integration.  All of the other pest management tactics work in combinations with either insecticides or natural enemies without any real problems.  We can’t blame lack of awareness of IPM as a cause for recurrent pest outbreaks.  Recurrent pest outbreaks are due to pesticides and cultural practices.

Lines 51-53  - Not having extension services is a major hindrance to successful pest management of any kind.  Again, IPM is a strategy to combine pest management tactics, focusing on successful use of natural enemies and not killing them off with insecticide applications during the cropping cycle.  IPM is not a tactic in and of itself.  If a small grower can successfully raise a crop with the judicious use of an insecticide and reduce pest reoccurrence through sound cultural practices, then that’s a win.  But, as the authors say, the lack of extension services to help guide growers to successfully use what tactics they have, no matter how limited they may be, is a hindrance to successful farming in the face of insect pest damage.

Lines 53-56.  I disagree with the use of the term integrated pest management here.  IPM is not a multifaceted approach, it is a strategy to ensure that natural enemies and chemicals can be successfully used together during a cropping cycle.  It is arguable that educating farmers is a part of IPM, but it is not in the strict definition.  It is however, inherent and crucial for success – again pointing to the importance of extension services.

Line 57 – IPM begins with accurate pest identification, not the life cycle, behavior or ecology.  That is a corner stone of IPM – accurate ID of the pest so then you can find out about its biology and then build a successful management program to control it.

Lines 59-60 are probably the two most important lines in the introduction.

Line 61 – equipping farmers with knowledge about how to use affordable and locally appropriate pest management techniques is crucial for successful implementation.  Talking about IPM in vague terms as if it were a management tactic and without specifics is the problem, it solves nothing. 

Lines 62 – 74 – It seems unnecessary to repeat the definition of IPM and the definitions of the different control tactics.  All of that is already well known and can simply be cited. 

Line 77 – Be careful with the descriptor “in depth”, that is a relative term that means different things to different people.  Let the people read your review and then decide for themselves if it’s in depth or not compared to other similar reviews.  It really isn’t for the authors to decide.  And saying an in-depth primer is an oxymoron.  A primer is a short introduction, usually meant for children learning something new, like how to read.  So, an in-depth primer is contradictory. 

Line 79 – based on the contents of the review, I would say that it does not focus on IPM.  And again, IPM is not a pest management tactic that reduces pest-related losses.  Insecticides, natural enemies, cultural practices or resistant host plants are what reduce pest-related losses.  How you combine these tactics during a cropping cycle is IPM.  And your review does not focus on how these tactics were integrated during a cropping cycle for any of the pests mentioned.

Line 80 - This review does not discuss the consequences of pesticide use to any significant degree, not as it pertains to specifically to rice production products and negative outcomes in SSA rice producing regions.

Line 89 – the paper needs to properly use scientific binomen styles for first use of a scientific name throughout.  (There are a lot that need fixing.)

Lines 91 and 92 have scientific names in different size font.

Line 112 – missing first bracket for citations

Line 142- 143 – needs citations

Lines 147 – 152 – no need to define crop rotation or cite a study on corn.

Lines 164 – 171 – no need to define intercropping.

Lines 184 – 194 – worth keeping in as this approach is not common in industrialized cropping settings.  Unfortunately, there are no examples listed with citations.  So, the impact, efficacy or practicality of this approach is not quantified.

Lines 196 – 202 – There’s often no distinction between a “weed” and a plant that is used for a cover crop, intercropping, insectary planting, etc.  Any plant that occurs in the cropping system in addition to the crop itself poses a risk of harboring unwanted pests.  Not just weeds.  And that is not necessarily a characteristic of weeds or defines what a weed is. 

Line 202 - should be the start of a new paragraph that summarizes the tactics described in Section 3.1 Cultural Practices.

Line 208 – do the authors mean their alternate host plants?

Line 210 – this section on resistance rice varieties is well done.

Line 263 – 264 – this sentence uses the term IPM accurately except that it is a singular framework, not plural.  Also the term IPM has already been introduced in the introduction so there is no need to spell out Integrated Pest Management here, IPM will suffice.

Lines 270 – 274 – no need to write out these already well known examples because they don’t pertain to rice.

Line 276 – Trichogramma is a genus name and should be italicized.

Line 280 – change to: “…the use of predatory beetles in the family Coccinellidae has…”

Lines 296 – 299 – the push-pull example from ICIPE is a cultural control method not biological control.  This should be moved to the appropriate section of the paper.

Line 304 – Biopesticides are a chemical control.  Biological control involves the use of living organisms only.  These should not be confused and the biopesticides should be moved to the chemical control section and table. Especially the neem-based products.  Those are in no way biological control.  It will be important for your paper to set and maintain correct terminology standards.

Line 319 – Tilapia nilotica should be italicized and the author name and family included in proper scientific binomen style for first use of a scientific name.

Line 337 – this is not a complete sentence. 

Line 344 is repetitive.

Line 348 – biological control is often not self-sustaining.  Introduction of new natural enemies in a classical biological control program requires a team of experts and tremendous capital inputs and infrastructure.  The cassava mealybug program is the case in point.  That is not something a grower does independently.  For rice production in SSA at the local level what growers are able to do is enhance already occurring natural enemies through conservation biological control (habitat enhancement and selective pesticides).  Independent growers are also able to use commercially available natural enemies for specific pest situations, like the ones mentioned in the table on biological control examples.  This method is referred to as augmentation biological control and is the most common practice of the three biological control methods.  The limitations for augmentation biological control is having access to the appropriate natural enemy when and where it is needed and in the correct amounts.  This is a challenge all over the globe. 

Line 355 – are fumigants actually a consideration? 

In addition – the section 3.5 Summary of the strength and limitations of cultural, biological, and chemical control is filled with generalities and no specifics regarding rice production in SSA.  It is far to generic, and all the pros and cons are well known already.  I’m not sure about the utility or need for this section unless it can somehow focus on rice production specifically.

Lines 370 – 408 – There is no need to redefine integrated pest management here.  The entire section is unnecessary as it covers basic textbook type definitions and examples that stray far away from the topic at hand.  This whole section should be removed.  Especially the IPM graphics.  They take up too much space and if they are examined closely the graphics do not actually show appropriate interactions for pest management in an applied situation like a rice farm in SSA.  In particular, the Dara 2019 graphic is practically useless as far as imparting understanding of IPM.  There are rings that don’t interconnect.  The outside ring suggests that economic viability influences environmental safety and that in turn influences social acceptability.  But it says that the reverse order doesn’t happen.  That economic viability has no impact on social acceptability.  And that none of these factors have any influence on the interior factors or producer, consumer or seller and that they have no impact on the outer ring.  It’s a mess.  Please remove these graphics.

Line 421 – does this section have rice examples?  Are drones with imaging technology available across SSA for individual growers to use and benefit from?  Keep this section if it can be tied directly to rice production in SSA with published examples. 

Line 431 – the same applies here – keep this section only if there are specific ties to rice production in SSA that are published.  This is a review paper of published works related to rice.

Line 445.  The authors have presented citations of examples of different pest management approaches for rice in SSA, but not really “case studies”.  For example, the push-pull technology was briefly mentioned and not presented as an in-depth case study and it was included in the wrong section of the paper.  Additionally, the push-pull program was developed decades ago, it was promoted for a brief time in the 1980s/1990s but has not been implemented beyond the ICIPE program because it is very case specific.  If it worked well and for other types of systems and could deliver return on investment, then growers would use it and it would be common with lots of examples.  But it doesn’t, so growers don’t use it, and it’s a nice idea, but not a significant contributor to effective, sustainable rice production in SSA.

Line 446 – 458 – something went wrong with the indentations.

Line 459 – 461 – this seems to be a crucial story to tell – if it pertains to rice production.  If it does, then this would be a good one to expand on as a true case study.

Line 469 – this information is basic and not needed.  There are no specific rice related examples. 

Line 481 – the section on gaps and future needs is far too generic.  Since this is a review paper on rice production, I question inclusion of this section unless it has specific rice-related gaps and future needs.

Line 528 -  Again, I would argue that the authors have not really presented case studies, but rather have cited a number of publications.

Review of sustainable pest management rice in Africa

Line 42 – there must be citations for the first sentence about rice as a crop.  If the authors use the term significant, then there needs to be sources to back up and provide data for why it is significant. 

Line 43 – Since this is a review and the authors are trying to establish the importance of insects as pests of rice, there needs to be a bit more explanation about it.  If insects are one of the major challenges, what are some examples of the other challenges?  If insects aren’t the most important challenge, then how do they rank?  Are they the second or third most important challenge? 

Line 48 – The term “prevalence” is confusing.  Do the authors mean how common the entire group of rice insect pests are in the entire area of SSA?  Prevalence is expressed as a percentage or rate out of some larger population or sample.  It could also have a time or seasonal factor.  High or low prevalence equates to widespread or rare, but that’s difficult to measure over such a vast area.  Do the authors mean localized prevalence?  In any case, insect species diversity – the number of different species in any given area and a species’ population abundance in any given area are influenced by an extraordinary number of factors beyond climate and cropping practices.  Things like commerce and artificial movement of species into an area, the floristic composition of an area, etc.  For any given farm, the types of insect pests and their abundances are influenced by surrounding farms, surrounding vegetation, weather and microclimates, and pest management practices in other managed areas.  The one thing that likely has little effect on the types of insect species over a wide area is availability of natural enemies.  Availability of natural enemies may not even have any impact on prevalence (the amount) of any given insect pest over a localized farmscape, let alone all of SSA, because availability doesn’t automatically translate to reductions in pest prevalence – in other words it is not a given. 

Line 49 – Are the authors referring to insect pests challenging rice production as the problem being exacerbated by a lack of effective pest management strategies, etc.?  The lack of awareness (by growers, I assume) of IPM principles and practices do not lead to recurrent pest outbreaks.  There needs to be a separation of pest management strategies and IPM.  In fact, I would use the term pest management tactics.  And how you group and use pest management tactics is the strategy.  IPM is not a pest management tactic.  IPM is an organizing principle – a strategy - for pest management tactics.  If there are pest management tactics that are missing or can’t be implemented, then the strategy is affected, and overall efficacy is compromised.  Having a lack of awareness of IPM is not that bad of an issue.  Growers will use whatever pest management tactics they have access to and can afford to try and manage insect pests.  The main question to ask is – are growers being thoughtful about using insecticides and natural enemies at the same time?  Because that is the crux of integrated pest management.  Those are the two tactics that need careful integration.  They are the only factors that need careful integration.  All of the other pest management tactics work in combinations with either insecticides or natural enemies without any real problems.  We can’t blame lack of awareness of IPM as a cause for recurrent pest outbreaks.  Recurrent pest outbreaks are due to pesticides and cultural practices.

Lines 51-53  - Not having extension services is a major hindrance to successful pest management of any kind.  Again, IPM is a strategy to combine pest management tactics, focusing on successful use of natural enemies and not killing them off with insecticide applications during the cropping cycle.  IPM is not a tactic in and of itself.  If a small grower can successfully raise a crop with the judicious use of an insecticide and reduce pest reoccurrence through sound cultural practices, then that’s a win.  But, as the authors say, the lack of extension services to help guide growers to successfully use what tactics they have, no matter how limited they may be, is a hindrance to successful farming in the face of insect pest damage.

Lines 53-56.  I disagree with the use of the term integrated pest management here.  IPM is not a multifaceted approach, it is a strategy to ensure that natural enemies and chemicals can be successfully used together during a cropping cycle.  It is arguable that educating farmers is a part of IPM, but it is not in the strict definition.  It is however, inherent and crucial for success – again pointing to the importance of extension services.

Line 57 – IPM begins with accurate pest identification, not the life cycle, behavior or ecology.  That is a corner stone of IPM – accurate ID of the pest so then you can find out about its biology and then build a successful management program to control it.

Lines 59-60 are probably the two most important lines in the introduction.

Line 61 – equipping farmers with knowledge about how to use affordable and locally appropriate pest management techniques is crucial for successful implementation.  Talking about IPM in vague terms as if it were a management tactic and without specifics is the problem, it solves nothing. 

Lines 62 – 74 – It seems unnecessary to repeat the definition of IPM and the definitions of the different control tactics.  All of that is already well known and can simply be cited. 

Line 77 – Be careful with the descriptor “in depth”, that is a relative term that means different things to different people.  Let the people read your review and then decide for themselves if it’s in depth or not compared to other similar reviews.  It really isn’t for the authors to decide.  And saying an in-depth primer is an oxymoron.  A primer is a short introduction, usually meant for children learning something new, like how to read.  So, an in-depth primer is contradictory. 

Line 79 – based on the contents of the review, I would say that it does not focus on IPM.  And again, IPM is not a pest management tactic that reduces pest-related losses.  Insecticides, natural enemies, cultural practices or resistant host plants are what reduce pest-related losses.  How you combine these tactics during a cropping cycle is IPM.  And your review does not focus on how these tactics were integrated during a cropping cycle for any of the pests mentioned.

Line 80 - This review does not discuss the consequences of pesticide use to any significant degree, not as it pertains to specifically to rice production products and negative outcomes in SSA rice producing regions.

Line 89 – the paper needs to properly use scientific binomen styles for first use of a scientific name throughout.  (There are a lot that need fixing.)

Lines 91 and 92 have scientific names in different size font.

Line 112 – missing first bracket for citations

Line 142- 143 – needs citations

Lines 147 – 152 – no need to define crop rotation or cite a study on corn.

Lines 164 – 171 – no need to define intercropping.

Lines 184 – 194 – worth keeping in as this approach is not common in industrialized cropping settings.  Unfortunately, there are no examples listed with citations.  So, the impact, efficacy or practicality of this approach is not quantified.

Lines 196 – 202 – There’s often no distinction between a “weed” and a plant that is used for a cover crop, intercropping, insectary planting, etc.  Any plant that occurs in the cropping system in addition to the crop itself poses a risk of harboring unwanted pests.  Not just weeds.  And that is not necessarily a characteristic of weeds or defines what a weed is. 

Line 202 - should be the start of a new paragraph that summarizes the tactics described in Section 3.1 Cultural Practices.

Line 208 – do the authors mean their alternate host plants?

Line 210 – this section on resistance rice varieties is well done.

Line 263 – 264 – this sentence uses the term IPM accurately except that it is a singular framework, not plural.  Also the term IPM has already been introduced in the introduction so there is no need to spell out Integrated Pest Management here, IPM will suffice.

Lines 270 – 274 – no need to write out these already well known examples because they don’t pertain to rice.

Line 276 – Trichogramma is a genus name and should be italicized.

Line 280 – change to: “…the use of predatory beetles in the family Coccinellidae has…”

Lines 296 – 299 – the push-pull example from ICIPE is a cultural control method not biological control.  This should be moved to the appropriate section of the paper.

Line 304 – Biopesticides are a chemical control.  Biological control involves the use of living organisms only.  These should not be confused and the biopesticides should be moved to the chemical control section and table. Especially the neem-based products.  Those are in no way biological control.  It will be important for your paper to set and maintain correct terminology standards.

Line 319 – Tilapia nilotica should be italicized and the author name and family included in proper scientific binomen style for first use of a scientific name.

Line 337 – this is not a complete sentence. 

Line 344 is repetitive.

Line 348 – biological control is often not self-sustaining.  Introduction of new natural enemies in a classical biological control program requires a team of experts and tremendous capital inputs and infrastructure.  The cassava mealybug program is the case in point.  That is not something a grower does independently.  For rice production in SSA at the local level what growers are able to do is enhance already occurring natural enemies through conservation biological control (habitat enhancement and selective pesticides).  Independent growers are also able to use commercially available natural enemies for specific pest situations, like the ones mentioned in the table on biological control examples.  This method is referred to as augmentation biological control and is the most common practice of the three biological control methods.  The limitations for augmentation biological control is having access to the appropriate natural enemy when and where it is needed and in the correct amounts.  This is a challenge all over the globe. 

Line 355 – are fumigants actually a consideration? 

In addition – the section 3.5 Summary of the strength and limitations of cultural, biological, and chemical control is filled with generalities and no specifics regarding rice production in SSA.  It is far to generic, and all the pros and cons are well known already.  I’m not sure about the utility or need for this section unless it can somehow focus on rice production specifically.

Lines 370 – 408 – There is no need to redefine integrated pest management here.  The entire section is unnecessary as it covers basic textbook type definitions and examples that stray far away from the topic at hand.  This whole section should be removed.  Especially the IPM graphics.  They take up too much space and if they are examined closely the graphics do not actually show appropriate interactions for pest management in an applied situation like a rice farm in SSA.  In particular, the Dara 2019 graphic is practically useless as far as imparting understanding of IPM.  There are rings that don’t interconnect.  The outside ring suggests that economic viability influences environmental safety and that in turn influences social acceptability.  But it says that the reverse order doesn’t happen.  That economic viability has no impact on social acceptability.  And that none of these factors have any influence on the interior factors or producer, consumer or seller and that they have no impact on the outer ring.  It’s a mess.  Please remove these graphics.

Line 421 – does this section have rice examples?  Are drones with imaging technology available across SSA for individual growers to use and benefit from?  Keep this section if it can be tied directly to rice production in SSA with published examples. 

Line 431 – the same applies here – keep this section only if there are specific ties to rice production in SSA that are published.  This is a review paper of published works related to rice.

Line 445.  The authors have presented citations of examples of different pest management approaches for rice in SSA, but not really “case studies”.  For example, the push-pull technology was briefly mentioned and not presented as an in-depth case study and it was included in the wrong section of the paper.  Additionally, the push-pull program was developed decades ago, it was promoted for a brief time in the 1980s/1990s but has not been implemented beyond the ICIPE program because it is very case specific.  If it worked well and for other types of systems and could deliver return on investment, then growers would use it and it would be common with lots of examples.  But it doesn’t, so growers don’t use it, and it’s a nice idea, but not a significant contributor to effective, sustainable rice production in SSA.

Line 446 – 458 – something went wrong with the indentations.

Line 459 – 461 – this seems to be a crucial story to tell – if it pertains to rice production.  If it does, then this would be a good one to expand on as a true case study.

Line 469 – this information is basic and not needed.  There are no specific rice related examples. 

Line 481 – the section on gaps and future needs is far too generic.  Since this is a review paper on rice production, I question inclusion of this section unless it has specific rice-related gaps and future needs.

Line 528 -  Again, I would argue that the authors have not really presented case studies, but rather have cited a number of publications.

Author Response

Dear Reviwer 2

Hope you are doing well?

Your comments help us improve our manuscript. Many thanks.

Please find attached the point-by-points answers to your comments.

Regards

Esther

Reviewer 3 Report

Comments and Suggestions for Authors

Suggestions and Edits:

Line 45: Include the scientific name of African rice gall midge (AfRGM).

Line 58-60: It would be good to include an example of this case here.

Line 88-89: Any source or data suggesting the estimated yield lose in numbers, giving an estimated percentage or number instead of using vague terms such as “substantial”.

Line 130: Correction- “armyworm”.

Line 140: Please include a paragraph (or along with table) of current management strategies (chemical control) for these pests’ management in different regions of Africa.

Line 145: What do you mean by “modern”?  

Line 149-152: Include the region for this example of crop rotation management.

Line 168-170: Provide more information about this example such as region and why did it improve crop resilience.

Line 180: “Synthetic pesticides” can be used instead of chemical pesticides throughout the review paper.

Line 211: It is not “plant diversity”, words like “plant breeding or genetics” can be used.

Line 296-299: Explain more “the push-pull strategy” and provide any examples and missing the citation.

Line 321: Which caterpillars and which crop/field?

Line 332-329: Any examples of agricultural pest control utilizing IFS approach.

Line 369: The examples of the IPM can be discussed in detail as this is the primary focus of this review paper.

Final suggestions: To include one paragraph highlighting the limitations/challenges of methods such as crop rotation, manual removal etc. with using these approaches in isolation, as well as other factors that may influence their effectiveness—similar to the discussion on the disadvantages of chemical control.

Author Response

Dear reviewer 3, 

Hope this mail find you in good health?

Your comment help us improve our manuscript. Many thanks.

Please find attached the point-by-point answers to your comments.

Regards

Esther 

Round 2

Reviewer 2 Report

Comments and Suggestions for Authors

Line 66: Rewrite as - ...practices, and the availability presence of natural enemies, influence the presence impact of insect pests.

Author Response

Dear Reviewer 2 , 

Hope this mail find you in good health.

Many thnaks for your comment. Please find attached the answer to your comment.  it's revised in yelow in the manuscript : see lines 70-72.

Do have a nice week ahead.

regards 
